# Comparison of the Analgesic Effect of Pericapsular Nerve Group Block and Lumbar Erector Spinae Plane Block in Elective Hip Surgery

**DOI:** 10.3390/medicina60050799

**Published:** 2024-05-11

**Authors:** Onur Küçük, Fatih Sağ, Ali Eyrice, Selman Karadayı, Ali Alagöz, Alkin Çolak

**Affiliations:** 1Department of Anesthesiology and Reanimation, Ankara Atatürk Sanatoryum Training and Research Hospital, University of Health Sciences, 06290 Ankara, Turkey; dr.okucuk@gmail.com (O.K.); mdalagoz@gmail.com (A.A.); 2Clinic of Anesthesiology and Reanimation, Tavşanlı Associate Professor Doctor Mustafa Kalemli State Hospital, 43300 Kütahya, Turkey; sag.fatih@gmail.com; 3Department of Anesthesiology and Reanimation, Trakya University Medical Faculty, 22030 Edirne, Turkey; alkincolak@trakya.edu.tr; 4Department of Anesthesiology and Reanimation, Kırklareli University Medical Faculty, 39100 Kırklareli, Turkey; selman.karadayi@klu.edu.tr

**Keywords:** hip, pericapsular nerve group block, lumbar erector spinae plane block, multimodal analgesia, pain postoperative

## Abstract

*Background and Objectives*: The aim of this study was to compare the effectiveness of pericapsular nerve group (PENG) and lumbar erector spinae plane (L-ESP) blocks, both administered with a high volume (40 mL) of local anesthetic (LA), for multimodal postoperative analgesia in patients undergoing hip surgery. *Materials and Methods*: This was a prospective, double-blind, randomized study that included 75 adult patients who were divided into three equal groups: control, PENG, and L-ESP. The study compared pain intensity, morphine consumption, time to first morphine request, and postoperative satisfaction between the control group, which received standard multimodal analgesia, and the block groups, which received PENG or L-ESP block in addition to multimodal analgesia. The numerical rating scale (NRS) was used to measure pain intensity. *Results*: The results showed that the block groups had lower pain intensity scores and morphine consumption, a longer time to the first morphine request, and higher postoperative satisfaction compared to the control group. The median maximum NRS score during the first 12 h was four in the control group, two in the PENG group, and three in the L-ESP group. The control group (21.52 ± 9.63 mg) consumed more morphine than the two block groups (PENG, 11.20 ± 7.55 mg; L-ESP, 12.88 ± 8.87 mg) and requested morphine 6.8 h earlier and 5 h earlier than the PENG and L-ESP groups, respectively. The control group (median 3) had the lowest Likert satisfaction scores, while the PENG group (median 4) had the lowest NRS scores (L-ESP, median 4). *Conclusions*: The application of PENG or L-ESP blocks with high-volume LA in patients undergoing hip surgery reduces the need for postoperative analgesia and improves the quality of multimodal analgesia.

## 1. Introduction

The hip joint connects the femur and pelvis and is responsible for multidimensional motion and mechanical stability [1]. Most hip pain is caused by the joint capsule [2]. Furthermore, the joint is largely controlled by the femoral, sciatic, and obturator nerves [3]. Patients may experience incomplete analgesia with techniques such as femoral nerve (FN) block alone due to the proximal locations of these nerves. Blocking the proximal innervation of all terminal branches innervating the hip joint (femoral, sciatic, obturator, etc.) causes significant weakness in the leg [1]. With the growth of the elderly population, more patients are undergoing hip surgeries. However, although methods are improving, no gold standard has been established for anesthesia or, more specifically, for analgesia in operations with high morbidity and high mortality [4].

Hip surgeries include hip arthroplasty (HA) (partial hip replacement, total hip replacement) and proximal femur operations (proximal femoral nailing) [5]. In fact, HA is one of the most successful orthopedic procedures in terms of improving patients’ functional status and quality of life [6]. Appropriate pain management for surgical patients contributes to early mobilization, shorter hospital stays, reduced costs, and increased patient satisfaction. Therefore, minimizing postoperative pain has become increasingly important for healthcare providers in recent years [7]. For this purpose, various analgesia techniques have been used. Although opioids typically provide effective pain relief, their use is limited due to serious side effects [8]. In recent years, peripheral nerve blocks (PNBs) have been used in the management of postoperative pain following hip surgery. The 2021 procedure-specific postoperative pain management (PROSPECT) guideline for total HA recommends several perioperative interventions to alleviate postoperative pain [9].

Erector spinae plane (ESP) block and pericapsular nerve group (PENG) block are the main safe and proven blocks used in hip operations [10,11]. PENG block is a novel regional analgesia technique that preserves motor function while reducing pain after HA. This technique involves injecting local anesthetic (LA) into the fascial plane between the psoas muscle and the superior pubic ramus [11,12]. ESP block is a paraspinal fascial plane block involving the injection of LA between the tip of the transverse process of the thoracic or lumbar vertebra and the anterior fascia of the erector spinae muscles [13]. The ESP block has primarily been used to provide analgesia in thoracoabdominal procedures as a potentially safer alternative to epidural or paravertebral techniques [14,15]. Its use in HA is limited in the literature, and its effectiveness and the amount of LA used are controversial [16,17].

The objective of this study was to compare the standard intravenous (IV) multimodal analgesia technique with ultrasound-guided (USG) PENG and ESP block techniques in postoperative analgesia management in hip surgery patients and to determine the most appropriate procedure for analgesia management.

## 2. Materials and Methods

### 2.1. Trial Design

This prospective, double-blind, randomized study was performed after the Trakya University Faculty of Medicine Scientific Research Ethics Committee approval (protocol number: TUTF-BAEK 2021/225; date: 17 May 2021). This study was registered with the ClinicalTrials.gov international protocol registration and results system under registration number NCT05802589. All procedures carried out in our study adhered to the ethical standards of the institutional and/or national research committee and the 1964 Declaration of Helsinki (as revised in 2013) and its subsequent amendments or comparable ethical standards.

### 2.2. Participants

Between May 2021 and December 2022, patients aged 18–80 years with American Society of Anesthesiologists (ASA) physical status classification scores of I–III who underwent elective hip or proximal femur surgery under general anesthesia participated in the study. The following were defined as exclusion criteria: refusal to consent, requesting exclusion from the study, allergy to LA drugs to be used in the study, infection at the site of intervention, body weight of less than 30 kg, dementia or cognitive impairment, bleeding disorders, liver failure, and chronic opioid or corticosteroid use. Additionally, patients with preoperative chronic pain were excluded from the study. Patients who met the inclusion criteria were included in the study after they provided written informed consent and underwent the application of a block for multimodal analgesia. For patient standardization, 4 patients with surgical procedures shorter than 60 min or longer than 180 min were excluded from follow-up. The Consolidated Standard of Reporting Trials (CONSORT) checklist, which was used for patient enrollment and distribution, is shown in Figure 1.

### 2.3. Interventions

#### 2.3.1. General Anesthesia Management

All patients undergoing elective hip or proximal femoral surgery in our clinic received the same anesthesia under the same protocol. The patients were premedicated with IV midazolam (0.03 mg/kg), and antibiotic prophylaxis was performed according to the hospital protocol. Anesthesia induction was performed with propofol (2–3 mg/kg), fentanyl (1.5 mcg/kg), and rocuronium bromide (0.7 mg/kg). For maintenance of anesthesia, a 0.6–0.8 minimum alveolar concentration (MAC) of sevoflurane and a 0.08 mcg/kg/min remifentanil infusion were used. Remifentanil dosing was adjusted according to hemodynamic parameters.

#### 2.3.2. Standard Analgesia Protocol

Our preoperative multimodal preemptive IV analgesia protocol included acetaminophen (1 g), dexketoprofen (50 mg/2 mL), and morphine (2 mg). All patients received the same postoperative multimodal analgesia with 1 g of IV acetaminophen three times daily. The patients were monitored postoperatively with a patient-controlled analgesia (PCA) device containing morphine. In the postoperative care unit (PACU), PCA was initiated when the patient reached a Ramsay Sedation Scale score of 3 or less and was programmed with a bolus of 2 mg morphine, with a 20 min lock time.

#### 2.3.3. Technique of PENG Block

The PENG block was administered in the supine position after general anesthesia and before the surgical incision, following the technique outlined by Girón-Arango et al. [11]. A curvilinear probe (Esaote, Europe B.V., Maastricht, The Netherlands) and a 20 G/80 mm needle (Pajunk, SonoPlex STIM, Geisingen, Germany) were used. The puncture was carried out in the lateromedial direction until the needle tip reached the plane between the iliopsoas tendon and periosteum and between the anterior inferior iliac spine (AIIS) and the iliopubic eminence (IPE) (refer to Figure 2(A1),(A2)). After localization was confirmed by hydrodissection (2 mL saline) to the plane under the iliopsoas muscle, 20 mL of 0.5% bupivacaine, 10 mL of 2% lidocaine, and 10 mL of normal saline (40 mL in total) were injected.

#### 2.3.4. Technique of L-ESP Block

The fourth lumbar vertebral level was determined using the conventional method (Tuffier’s line, an imaginary line between the two iliac crests). A curvilinear probe (Esaote, Europe B.V., Maastricht, The Netherlands) and a 20 G/80 mm needle (Pajunk, SonoPlex STIM, Geisingen, Germany) were used. The curvilinear USG probe was placed over the spinous processes on the midvertebral line in the sagittal plane. The transducer was then moved 3.5–4 cm laterally from the midline to visualize the erector spinae muscle and transverse processes. Using the in-plane technique, a puncture was performed until the block needle reached the transverse process (refer to Figure 2(B1),(B2)). After confirming the location by hydrodissection (2 mL saline) of the plane under the erector spinae muscle, 20 mL bupivacaine 0.5%, 10 mL lidocaine 2%, and 10 mL normal saline (40 mL total) were injected.

#### 2.3.5. Assessment of Pain

The Numeric Rating Scale (NRS) was used to assess postoperative pain. Pain was measured at a total of 6 time points (T0, T1, T2, T3, T4, T5, and T6). T0 was when the patients had no pain during the block procedure under general anesthesia. The initial measurement point (T1) was taken in the post-anesthesia care unit (PACU) when the patient achieved a Ramsay Sedation Scale score of 3 or lower. This measurement was taken within the first hour following the operation. The time point T2 occurred 2 h after surgery. The NRS values evaluated 4, 6, 12, and 24 h later were the T3, T4, T5, and T6 values, respectively.

#### 2.3.6. Evaluation of Satisfaction

A 5-point Likert satisfaction scale was used to measure postoperative patient satisfaction. At the end of the 24th postoperative hour, the patients were asked to choose 1 of 5 options describing the last 24 h: not at all satisfied (1 point), dissatisfied (2 points), neither dissatisfied nor satisfied (3 points), satisfied (4 points), or very satisfied (5 points).

### 2.4. Outcomes

The study’s primary outcome measure was the NRS pain scores recorded at rest 4 h postoperatively. The secondary outcome measures included the NRS pain scores at rest during other time points, patient analgesic requirement measured by morphine consumption through PCA, and patient satisfaction score on a 5-point Likert scale 24 h post-operation.

### 2.5. Sample Size

To determine the required sample size, we assessed our primary hypothesis that adding PENG or L-ESPB block to the standard analgesia procedure in multimodal analgesia management would enhance postoperative analgesia. Comparisons between the three groups are rare in the literature. Tulgar et al. compared the control group with two peripheral blocks (L-ESPB and Quadratus Lumborum Block) and used the NRS to assess pain [10]. Upon examination of the study results, it was found that the average NRS values of the control group at the 3rd hour (when the effect of perioperative analgesia decreased) were 2.00 ± 0.46, while those of the L-ESPB and QLB groups were 1.45 ± 0.51 and 1.00 ± 0.65, respectively. Based on these results, the study’s effect size was 1.13. When examining literature studies that use PENG block for postoperative analgesia management in hip surgery and evaluate pain with NRS, the average 24 h pain score in the block group was found to be 2 [18,19,20]. This value was similar to the 24 h results of the groups that received the block in the study conducted by Tulgar et al.

The sample size of this study was calculated using G*Power software version 3.1.9.6 (Institute of Experimental Psychology, Heinrich Heine University, Düsseldorf, Germany), taking into account the aforementioned calculations. The sample size was determined to be 18 people per group, with a power of 0.90, a significance level of 0.05, and an effect size of 1.13. In order to reduce the margin of secondary error, we designed our study to include 25 patients in each group, and a total of 75 patients were planned to be included in our study consisting of 3 groups.

### 2.6. Randomization

This study involved 3 groups: a PENG group, a L-ESPB group, and a control group (without block). Each group comprised 25 patients. Each patient was randomly assigned an ID number before surgery, and all data were collected using this ID number. A randomization table was used to assign the patients to the 3 groups. When each patient arrived, the physician who performed the randomization informed the physician who would perform the procedure about whether to use a block and, if so, which one to perform. The blocks were performed by an anesthesiologist experienced in the application and who was not involved in either randomization or data collection. All data were collected blindly by a physician other than the one who performed the randomization and administered the block. The patients underwent follow-up after the surgeons and anesthesiologists had completed their procedures. Physicians’ choices regarding the procedure were not interfered with. The patients were informed of the use of the PCA device in detail following surgery, and the administration of drugs was not impeded. The patients were instructed to press the button provided to them on their hand in the event of experiencing pain. The PCA devices were collected 24 h after the operation by the data-collecting physician. The first morphine administration, morphine consumption, and total demand amounts were obtained from the device records.

The occurrence of postoperative complications and side effects such as hypotension, allergic reactions, neurological complications, respiratory depression, sedation, urinary retention, nausea or vomiting, and infection were recorded. Recorded patient data included age, height, body weight, body mass index (BMI), gender, ASA physical status, operation type, operation time, postoperative NRS scores, first morphine request time, total morphine consumption, and patient satisfaction. Data on the PCA device provided information on the time to the first morphine request and the total morphine consumption.

### 2.7. Statistical Analysis

All data obtained during the study and recorded in the study form were analyzed using the Jamovi statistical program, version 2.3.21.0 (Sydney, Australia), and we created graphical representations. After evaluating whether the data were normally distributed, we used the Shapiro–Wilk test, histograms, and Q–Q plots to express the mean ± SD for normally distributed data, and we used median quartiles for non-normally distributed or ordinal data. Categorical variables are presented in terms of the number and percentage of cases, and we evaluated them using chi-squared and Fisher’s exact tests. Because this study involved 3 groups, we analyzed continuous variables using Welch’s and Fisher’s one-way ANOVAs or the Kruskal–Wallis test. Differences between groups were analyzed with the post hoc Tukey test, the Games–Howell post hoc test, or the post hoc Dwass–Steel–Critchlow–Fligner (DSCF) test. When appropriate, we calculated 95% confidence intervals (CIs), and we considered *p*-values of less than 0.05 to be statistically significant.

Significance values were adjusted with the Bonferroni correction for multiple comparisons (NRS scores), and when comparing 3 groups, we considered *p*-values below 0.016 to be statistically significant.

## 3. Results

In total, 88 patients undergoing hip or proximal femur surgery were assessed for inclusion in this study. Six patients were excluded because of dementia and/or cognitive impairment, one patient was excluded because of chronic opioid use, and two patients were excluded for chronic corticosteroid use. Additionally, four patients were excluded because their surgical procedures exceeded 180 min in duration. Table 1 includes age, gender distribution, BMI, ASA physical status, and operation type and duration data. The three groups demonstrated no statistically significant differences in terms of these parameters. The median pain scores of the groups at rest in the operating room before the operation were 1 (0–2) in the control group, 1 (0–1) in the PENG group, and 1 (0–2) in the L-ESP group, with no statistically significant difference (Kruskal–Wallis, *p* = 0.901).

Table 2 shows the patients’ resting NRS scores at the various time points; the groups demonstrated statistically significant differences at T1, T2, T3, and T4 (Kruskal–Wallis, *p* = 0.019, *p* = 0.002, *p* < 0.001, *p* < 0.001). At T5 and T6, the groups demonstrated no statistically significant differences in NRS scores (Kruskal–Wallis, *p* = 0.559, *p* = 0.943). Figure 3 is a graph of the resting NRS standard error among the groups at each time point.

The difference between the control group and the PENG group was the only reason for the statistically significant difference at time point T1 (post hoc DSCF, *p* = 0.006). At time points T2 (post hoc DSCF, *p* = 0.001, *p* = 0.012), T3 (post hoc DSCF, *p* < 0.001, *p* = 0.001), and T4 (post hoc DSCF, *p* < 0.001, *p* = 0.002), the statistically significant differences were due to the difference between the control group and both other groups. The NRS score of the control group was statistically higher than the other two groups at the T2, T3, and T4 time points. There was no statistically significant difference observed between the PENG and L-ESP groups in terms of the NRS score at T1, T2, T3, and T4 time points.

Comparing the groups in terms of their maximum NRS scores during the first 12 h revealed a statistically significant difference (Kruskal–Wallis, *p* < 0.001). The control group’s median maximum NRS score in the first 12 h was higher than those of the other two groups (post hoc DSCF, *p* < 0.001, *p* = 0.001). No statistically significant difference between the PENG and L-ESP groups emerged in terms of maximum NRS scores in the first 12 h (post hoc DSCF, *p* = 0.292).

The groups demonstrated significant differences in terms of postoperative morphine consumption (Fisher’s ANOVA, *p* ≤ 0.001, Table 2, Figure 4). The mean morphine consumption in the control group was 21.5 mg (SD = 9.6 mg), which was 10.3 mg more than that in the PENG group (mean = 11.2 mg, SD = 7.5 mg, *p* ≤ 0.001) and 8.6 mg more than that in the L-ESP group (mean = 12.8 mg, SD = 8.8 mg, *p* ≤ 0.001). The differences between the control group and both other groups were statistically significant (post hoc Tukey’s test). No statistically significant difference was observed between the PENG and L-ESP groups (*p* = 0.258).

The groups’ first postoperative morphine demand times differed significantly (Welch’s ANOVA, *p* ≤ 0.001, Table 2). The mean time to the first morphine demand in the control group was 3 h (SD = 2.1 h), which was 6.8 h earlier than that in the PENG group (mean = 9.8 h, SD = 6.3 h, *p* ≤ 0.001) and 5 h earlier than that in the L-ESP group (mean = 8 h, SD = 5.6 h, *p* ≤ 0.001). The differences between the control group and both other groups were statistically significant (post hoc Games–Howell). No statistically significant difference was observed between the PENG and L-ESP groups (*p* = 0.204).

Table 2 compares the groups’ Likert satisfaction scores and side effects at the end of the 24th postoperative hour. The median Likert satisfaction score in the control group at the end of the 24th hour was 3 (2–4), lower than those in the other two groups, and this difference was statistically significant (post hoc DSCF, *p* < 0.001, *p* = 0.006). No statistically significant difference was observed between the PENG and L-ESP groups in terms of Likert satisfaction scores after 24 h (*p* = 0.233).

Finally, there were no significant differences in the incidence of side effects between the groups. Nausea was observed in seven patients, vomiting in four, and dizziness in two (*p* = 0.332, Table 2).

## 4. Discussion

The study results indicate that both the USG PENG block, applied to the plane under the iliopsoas muscle with 40 mL of LA mixture, and the L-ESP block, performed at the level of the 4th lumbar vertebra, produce effective postoperative analgesia after hip surgery when compared to the standard multimodal IV analgesia regimen.

Improvements in surgical procedures and increasing compliance with Enhanced Recovery After Surgery (ERAS) protocols have a positive impact on patient outcomes [21]. In 2020, the ERAS^®^ association published recommendations for hip surgery, emphasizing the importance of an opioid-sparing multimodal analgesic approach and early mobilization [22]. Therefore, effective multimodal analgesic management plays a crucial role in success. Peripheral fascial blocks are particularly important for this purpose.

One of these, the PENG block was first applied successfully in 2018 by Girón-Arango et al. [11] with 20 mL LA in five patients for postoperative analgesia in hip surgeries. In the first applications, it was thought that 20 mL LA would only affect the nerve branches providing pericapsular analgesia, and wound LA infiltration was added to the block to provide surgical site dermatome analgesia [18]. On the other hand, some studies have found that the PENG block has a wider area of effect with 20 mL LA [23,24]. Although LA is applied as 3 mL/kg (maximum 40 mL) in peripheral plan blocks [25], there are no clinical studies involving the use of 40 mL LA in PENG block. In cadaveric studies in which different volumes were used, very different LA distributions were reported [26,27]. In addition, different analgesic results have been reported in some case reports using up to 40 mL of LA [28]. A cadaveric study by Yamak et al. using different volumes of LA (5 to 20 mL, 0.5% bupivacaine) showed that it was possible to reach the lateral femoral cutaneous nerve and FN with high-volume LA application in the region between the psoas tendon and bone tissue [26]. Moreover, the LA spread to the posterior part of the hip and covered part of the sciatic nerve. Considering the partial sciatic nerve involvement seen in different LA volumes and the fact that the volume limit may go up to 3 mL/kg (maximum 40 mL) in fascial blocks, we used 40 mL of 0.25% bupivacaine in PENG. The present study is the first prospective randomized study in the literature in which this was completed. The results of this study demonstrate the significant analgesic effectiveness of the PENG block compared to standard multimodal analgesia. Additionally, although not statistically significant, NRS scores were lower in the PENG group than in the L-ESP block group at every time point. This was also observed in the amount of morphine consumed. At the same time, the first morphine demand was 9.8 h (SD = 6.3) in the PENG group, whereas it was 3 h in the standard IV analgesia group. Notably, the use of plane blocks in conjunction with multimodal analgesia resulted in better Likert scores. These findings demonstrate that a combination of high-volume PENG block or L-ESP block with standard IV multimodal analgesia can effectively manage pain and improve patient satisfaction. Furthermore, the limited consumption of opioids may aid in the early recovery of patients by reducing the adverse effects of opioids during the postoperative period. In most studies in the literature, LA is applied for wound infiltration, according to the idea that PENG block provides only pericapsular analgesia [29]. This study found that providing effective analgesia is possible with PENG block applied with a high volume, without the need for wound infiltration. The results also suggest that PENG block applied with a high volume has a similar effect to lumbar plexus block.

The basic ESP block technique involves USG injection of a large volume of LA (0.3–0.5 mL/kg) into the fascial plane between the ends of the vertebral transverse processes and the erector spinae muscle [13]. LA spreads craniocaudally to 3–6 vertebral levels within this potential space. Lumbar imaging studies on ESP have shown it to spread to the lumbar plexus [30,31]. In a study on hip surgery, Tulgar et al. [10] applied an L-ESP block with 40 mL LA to patients under general anesthesia and found a statistically significant reduction in patients’ NRS scores compared to those achieved with standard multimodal IV analgesia from the first hour postoperatively. In this study, similar results were obtained with Tulgar et al. However, the L-ESP group showed significantly more effective analgesic results compared to standard IV multimodal analgesia. Additionally, while similar results were obtained with the PENG block, the analgesic efficacy and patient satisfaction were better with the PENG block. Based on these results, it can be concluded that L-ESP can produce an effect similar to the lumbar plexus block. The application of L-ESP block may be impeded by difficulties in positioning in awake patients. However, we did not encounter such a problem as we performed our applications under general anesthesia.

The 2016 guidelines of the American Pain Society, the American Society of Regional Anesthesia and Pain Medicine, and the American Society of Anesthesiologists recommend that safe and effective postoperative pain management should be tailored to each patient and to the relevant surgical procedure in addition to recommending multimodal analgesia procedures [7]. The major benefit of applying regional anesthesia techniques in multimodal analgesia procedures is the reduced consumption of opioid analgesic drugs [32]. Regional anesthesia techniques are highly valuable for reducing opioid use in postoperative pain control and even for providing opioid-free pain management. In the present study, we evaluated PENG block and L-ESP block in terms of their effects on opioid use. Morphine consumption was lower in both block groups than in the control group. According to these results, both blocks (PENG and L-ESP) reduced postoperative opioid consumption.

Regional anesthesia techniques are part of multimodal analgesia in hip surgery [33]. Although epidural analgesia is the gold standard in patients undergoing hip surgery, other options that provide effective analgesia include quadratus lumborum block, paravertebral block, psoas compartment block, and transverse abdominis plane block [33,34]. These blocks applied in hip surgery, especially epidural analgesia, are methods that should be performed with caution and only by experienced physicians due to serious side effects and complications. Regarding complications after PENG block, only two cases of quadriceps motor weakness have been reported to date [23]. A review of 45 randomized trials involving thoracic ESP blocks (1904 blocks in 1386 patients) reported no complications [35]. In our study, no block-related complications developed in any of the patients. The fact that these two blocks do not have direct vascular, nerve, or organ contact suggests that they may be safer than other techniques used in hip surgery.

The most significant challenge in regional anesthesia techniques for postoperative analgesia is block failure. Although the use of USG in PNB applications has been shown to reduce the incidence of failure due to the enhanced visualization of the nerve, plexus, or fascial plane, a study by Sites et al. [36] in 2007 found that USG-guided nerve blocks performed by trainees had a block failure rate of approximately 6.4%. Furthermore, block failure, technical and surgical factors that may impair LA propagation, patient-induced position, and anatomical variations may affect the success of PNBs [37]. A comprehensive study of over 7000 patients revealed an overall success rate of 89% following PNB application [38]. This indicates that approximately 1 in 10 PNBs is not effective. L-ESP and PENG blocks are fascial plane blocks. In this regard, the success rate of plane blocks has been found to be higher than that of specific nerve blocks [37]. It is also known that the administration of regional anesthesia can cause pain and anxiety in the patient during the procedure [39]. In the study published by Kessler et al. [40] in 2013, the risk assessment of paresthesia, injection pain, and nerve damage in peripheral regional anesthesia applied to patients under general anesthesia and conscious patients was compared. It was found that paresthesia and nerve pain were associated with neurological deficits, although this situation was detected earlier in awake patients who were lightly sedated. However, no scientific evidence was found in the literature to support this finding. Furthermore, there was no discernible difference between the two groups in terms of the risk of local anesthetic poisoning [40]. In light of the aforementioned literature data, there is no discernible difference in terms of block success and the incidence of side effects in patients undergoing peripheral block application under light sedation or general anesthesia. In the course of our study, we proceeded to perform the block application subsequent to the administration of general anesthesia. Thus, the objective was to eliminate the pain and anxiety experienced by patients during the block application. By eliminating the existing baseline pain, it was ensured that the postoperative pain comparison between the groups was more standardized. However, it should be noted that a dermatomal block performance analysis was not performed, as the block application was conducted under general anesthesia. It is our opinion that further studies should be conducted on this subject.

Our study had several limitations. Firstly, we compared resting NRS pain scores in three groups at six measurement points. Although the mixed ANOVA test would have provided a stronger evaluation in the results section, we could not apply it due to the categorical ordinal variable NRS being used in our study and our measured values not complying with normal distribution. Measurements were compared separately between the three groups at six time points, and the maximum NRS value in the first 12 h was evaluated. It was not possible to evaluate preoperative NRS pain scores separately, as each patient received perioperative standard analgesia and all patients underwent general anesthesia, thus eliminating the existing baseline pain. Furthermore, given that both blocks were conducted under general anesthesia, it was not possible to perform the requisite examinations to evaluate the effectiveness of the blocks, namely, dermatomal analysis or pain score evaluation after block application. Secondly, the NRS scores were higher in the control group than in the peripheral block groups at various time periods. It is possible that this difference is due to the inadequacy of the multimodal analgesia regimen. However, it is important to note that the goal was to reduce patients’ opioid consumption. Thirdly, it was not possible to conduct a detailed evaluation of whether the applied blocks resulted in postoperative loss of muscle strength. Although we examined the pain scores of the patients on movement in the PACU, our orthopedic physicians require that patients undergoing hip surgery remain immobile for the first 12–24 h postoperatively in accordance with their clinical procedures. Consequently, an effective muscle strength examination could not be performed. Nevertheless, it is important to note that the application of PENG and L-ESP blocks with high LA volumes (40 mL) may result in postoperative muscle weakness. This issue should be taken into consideration. Further studies are required to address this issue. Additionally, although surgical procedures were performed using a standardized technique from the practicing clinic, they were not all performed by the same surgeon. Therefore, the patients’ varying procedures may have caused different areas and intensities of pain. 

Finally, we examined the analgesic efficacy of the groups that received PENG block and ESP block, compared to the control group. The sample size for our study was calculated accordingly. Due to the small sample size, there was a high secondary margin of error in the comparisons between the PENG and ESP groups.

## 5. Conclusions

Regional anesthesia techniques are an important part of postoperative multimodal analgesia management. In patients undergoing hip surgery, PENG block or L-ESP block applied with high-volume LA (20 mL bupivacaine 0.5%, 10 mL lidocaine 2%, and 10 mL normal saline) reduces postoperative patients’ analgesia requirements and improves the quality of multimodal analgesia. Our study is the first prospective study in the literature in which high-volume LA was applied in PENG block. PENG block applied with a 40 mL LA volume has a wide range of potential effects in the hip region and may have a lumbar plexus block-like effect. Studies are needed to determine the most effective LA volume and concentration of the two blocks to use for lower extremity hip surgery, while avoiding muscle weakness.

The findings of this study indicate that both blocks significantly enhanced the quality of multimodal analgesia following hip surgery and markedly reduced postoperative opioid consumption. There is no discernible difference between the two blocks in terms of their efficacy in postoperative pain management. Both blocks are plane blocks, and the risk of complications is low due to the absence of vascular, nerve, and organ contact. These two techniques can be safely applied with high-volume LA in hip surgery.

## Figures and Tables

**Figure 1 medicina-60-00799-f001:**
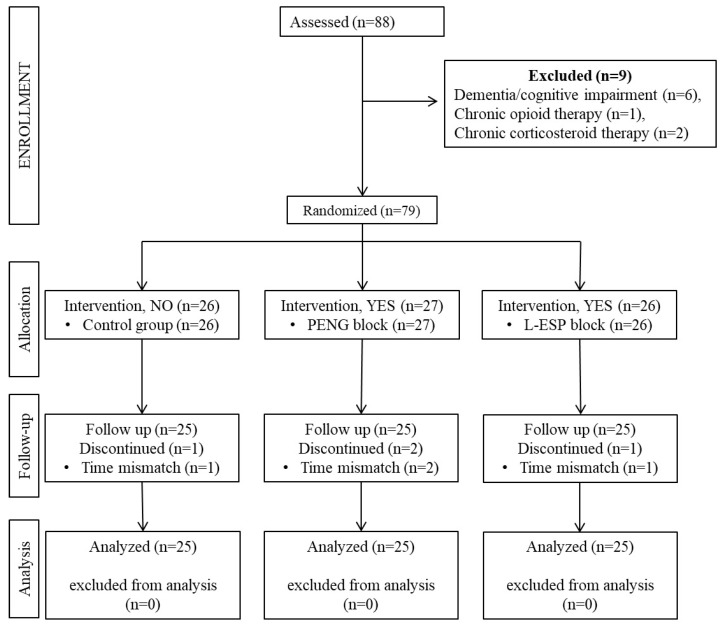
CONSORT flow diagram of the study population (PENG: pericapsular nerve group, L-ESP: lumbar erector spinae plane).

**Figure 2 medicina-60-00799-f002:**
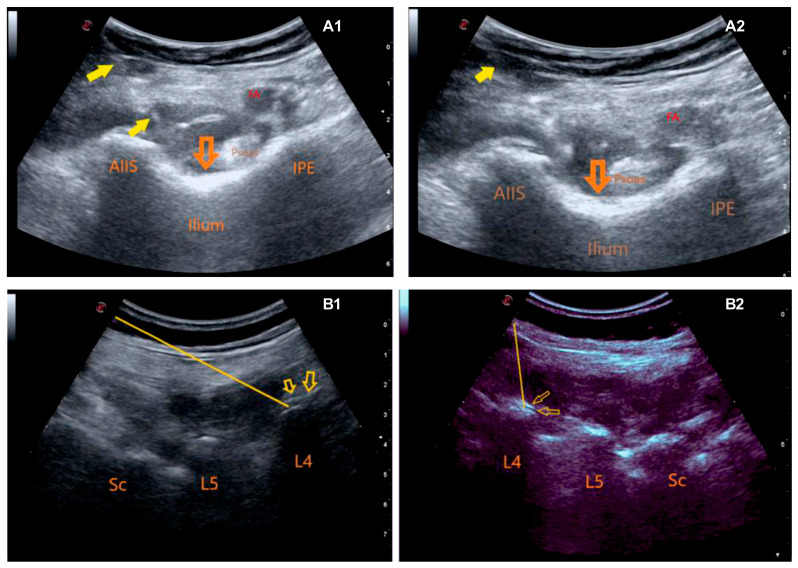
Pericapsular nerve group block and lumbar erector spinae plane block application techniques: (**A1**,**A2**) Pericapsular nerve group block application (FA: femoral artery, AIIS: anterior-inferior iliac spine, IPE: iliopubic eminence, Yellow arrow: injection needle, Orange arrow: local anesthetic area); (**B1**,**B2**) Lumbar erector spinae plane block application (L4: Transverse process of the 4th lumbar vertebra, L5: Transverse process of the 5th lumbar vertebra, Sc: Sacrum, Yellow line: injection needle, Yellow arrow: local anesthetic area).

**Figure 3 medicina-60-00799-f003:**
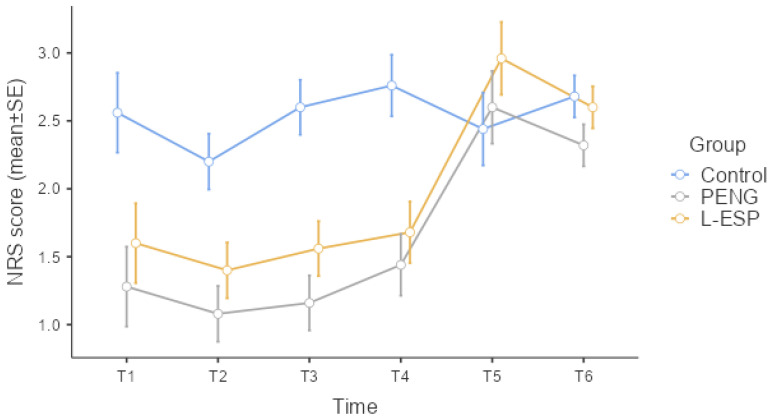
NRS score standard error graph at time points according to groups (NRS: numeric rating scale, PENG: pericapsular nerve group, L-ESP: lumbar erector spinae plane, SE: standard error).

**Figure 4 medicina-60-00799-f004:**
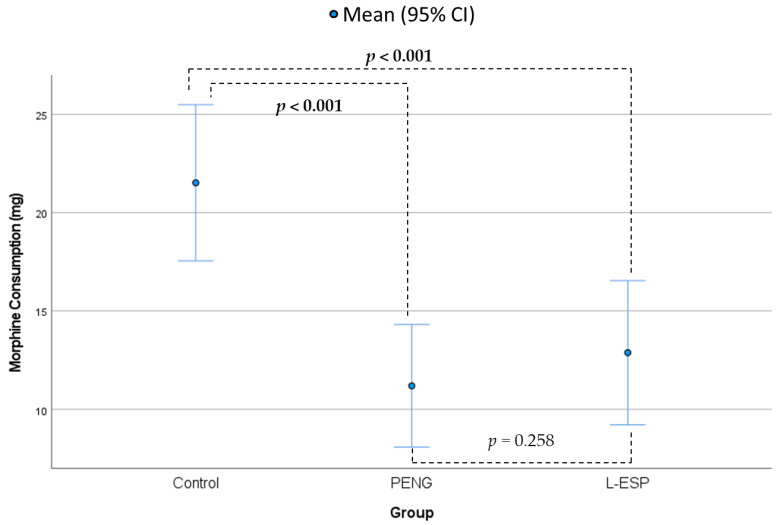
Postoperative morphine consumption according to groups (PENG: pericapsular nerve group, L-ESP: lumbar erector spinae plane, CI: confidence interval).

**Table 1 medicina-60-00799-t001:** Demographic and surgical information by groups.

	Control (*n* = 25)	PENG (*n* = 25)	L-ESP (*n* = 25)	*p* Value
Age (year)	65.88 ± 11.44	64.40 ± 11.34	64.96 ± 10.69	0.967 *
Sex, *n* (%)				1.000 ^†^
Female	18 (72%)	18 (72%)	18 (72%)	
Male	7 (28%)	7 (28%)	7 (28%)	
BMI (kg/m^2^)	25.9 ± 4.24	27.2 ± 4.14	26.7 ± 4.86	0.491 *
ASA, *n* (%)				0.442 ^†^
1	-	2 (8%)	2 (8%)	
2	13 (52%)	13 (52%)	16 (64%)	
3	12 (48%)	10 (40%)	7 (28%)	
Operation time (minute)	147 ± 22.8	145 ± 24.8	148 ± 29.7	0.881 *
Operation type, *n* (%)				0.997 ^†^
Partial hip replacement	10 (40%)	11 (44%)	10 (40%)	
Totally hip replacement	7 (28%)	7 (28%)	7 (28%)	
Proximal femoral nailing (PFNA)	8 (32%)	7 (28%)	8 (32%)	

Continuous variables are expressed as either the mean ± standard deviation (SD), and categorical variables are expressed as either frequency (*n*) or percentage (%). Continuous variables were compared with Welch’s and Fisher’s One-Way ANOVA tests or the Kruskal–Wallis test *. Categorical variables were compared using Pearson’s chi-square test or Fisher’s exact test ^†^. BMI: Body mass index. ASA: American Society of Anesthesiologists.

**Table 2 medicina-60-00799-t002:** NRS scores, morphine demands, morphine consumption, Likert satisfaction scales, and side effects by groups.

	Control (*n* = 25)	PENG (*n* = 25)	L-ESP (*n* = 25)	*p* Value
0–12 h maximum NRS score, median (Q1–Q3)	4 (3–4)	2 (2–4)	3 (2–4)	**<0.001** ^a,^*
NRS, T1, median (Q1–Q3)	2 (1–3)	1 (0–2)	1 (1–2)	**0.019** ^b,^*
NRS, T2, median (Q1–Q3)	2 (1–3)	1 (0–2)	2 (1–2)	**0.002** ^a,^*
NRS, T3, median (Q1–Q3)	2 (2–3)	1 (0–2)	2 (1–2)	**<0.001** ^a,^*
NRS, T4, median (Q1–Q3)	2 (2–4)	1 (1–2)	2 (1–2)	**<0.001** ^a,^*
NRS, T5, median (Q1–Q3)	2 (2–3)	2 (2–3)	3 (2–4)	0.559 *
NRS, T6, median (Q1–Q3)	2 (2–3)	2 (2–3)	2 (2–3)	0.943 *
Morphine request postoperatively, *n* (%)				0.070 ^†^
Yes	25 (%100)	20 (%80)	22 (%88)	
No	0	5 (%20)	3 (%12)	
Total morphine consumption (mg), mean ± SD	21.52 ± 9.63	11.20 ± 7.55	12.88 ± 8.87	**<0.001** ^a,^*
First morphine demand time, hour	3.0 ± 2.1	9.8 ± 6.3	8.0 ± 5.6	**<0.001** ^a,^*
24 h Likert scale, median (Q1–Q3)	3 (2–4)	4 (4–5)	4 (3–5)	**<0.001** ^a,^*
1, *n* (%)	3 (%12)	0	1 (%4)	
2, *n* (%)	4 (%16)	2 (%8)	2 (%8)	
3, *n* (%)	10 (%40)	4 (%16)	7 (%28)	
4, *n* (%)	8 (%32)	9 (%36)	6 (%24)	
5, *n* (%)	0	10 (%40)	9 (%36)	
Adverse effects, *n* (%)				0.332 ^†^
Yes (nausea, vomiting, dizziness)	4 (%16)	1 (%4)	2 (%8)	
No	21 (%84)	24 (%96)	23 (%92)	

^a^ The statistically significant difference is due to the difference between the control group and other groups. ^b^ The statistically significant difference is due to the difference between the PENG group and the control group. Continuous variables are expressed as the mean ± standard deviation (SD) or median (quartiles) and categorical variables are expressed as either frequency (*n*) or percentage (%). Continuous variables were compared with Welch’s and Fisher’s One-Way ANOVA tests or the Kruskal Wallis test *. Categorical variables were compared using Pearson’s chi-square test or Fisher’s exact test ^†^. Statistically significant *p*-values are in bold. *p*-Values of less than 0.05 to be statistically significant. When comparing the 3 groups, Bonferroni correction was applied. NRS: numerical rating scale.

## Data Availability

The datasets used and/or analyzed during the current study available from the corresponding author on reasonable request.

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
