# Peer review of "Comparison of the Analgesic Effect of Pericapsular Nerve Group Block and Lumbar Erector Spinae Plane Block in Elective Hip Surgery"

_medicina, 2024, doi:10.3390/medicina60050799_

Round 1

Reviewer 1 Report

Comments and Suggestions for Authors

Kücük et al investigate whether the PENG Block and erector spinae block are superior to standard analgesic treatment in hip surgery. The authors have tob e congratulated for their work which adds more evidence for regional anaesthesia in hip surgery. The manuscript is well written and I have only minor comments:

Minor:

Abstract:

Please provide some numbers for comparison of the groups

Results:

-          How did the authors test for effectiveness of the block?

-          Authors should provide numbers from resting paing before surgery

-          The authors sshould also provide data about muscle weakness or mobility since many surgeons require rapid recovery of muscle strength after surgery and early mobilization

Discussion:

-          While the blocks were performed under general anaesthesia it is not possible to evaluate the effectiveness oft he block. This should be included in the discussion section and should be discussed in the light oft he recent literature.

Author Response

Dear Editor,

We thank the reviewers for their valuable comments on the manuscript, and we have edited the manuscript to address their concerns. We have highlighted in red all the changed places in the main text. We would like to state that all the reviewers' comments have been taken into account and the necessary corrections have been made.

We hope that the article is now suitable for publication.

Yours sincerely

Ali EYRÄ°CE, MD.

Department of Anesthesiology and Reanimation, Trakya University Medical Faculty, Edirne, Turkey

On behalf of all authors.

Reply to Reviewer-1:

Kücük et al investigate whether the PENG Block and erector spinae block are superior to standard analgesic treatment in hip surgery. The authors have tob e congratulated for their work which adds more evidence for regional anaesthesia in hip surgery. The manuscript is well written and I have only minor comments:

Minor:

Abstract:

Please provide some numbers for comparison of the groups

 A-1) I would like to express my gratitude for your invaluable contribution. In accordance with your recommendation, the conclusion section of our summary has been expanded, and the results of the groups are indicated with numbers.

Results:

-          How did the authors test for effectiveness of the block?

A-2) I would like to express my gratitude for your invaluable contribution. All blocks were performed under ultrasound guidance by the same experienced and active anaesthetist as previously stated in the material and method section of our study. It should be noted that the blocks performed were not specialised blocks, but rather plan blocks. It should be noted that a separate paragraph has been added to the discussion section to address this point. As all of the blocks were performed under general anaesthesia, there was no opportunity to test or perform a dermatomal examination following the application of the blocks. It should be noted that this situation has been previously disclosed in the limitation section of the study, as presented in the initial submission. In order to clarify this point, we have expanded the limitations section of our study to provide a more detailed account of the circumstances that led to this misunderstanding. Conversely, the statistically significant difference between the control group and the block group in NRS pain scores evaluated in the PACU unit at the T1 time point in our results clearly demonstrates the effectiveness of the blockade. We would like to express our gratitude for your insightful comments and assessments.

-          Authors should provide numbers from resting paing before surgery

A-3) Firstly, I would like to express my gratitude for your invaluable input. It should be noted that our hospital has a standardised protocol for the administration of analgesia prior to surgery, which is outlined in the Material and Methods section of our inaugural article. Furthermore, it should be noted that special attention is paid to the absence of pain in our preoperative patients, who are recorded for the study. As the initial version of our article included a standard preoperative analgesia protocol, we did not present the preoperative pain scores. This was also stated in the limitations section of the study. However, in response to your valuable request, we have included the NRS pain scores between the groups evaluated preoperatively in the operating theatre in the results section. We are grateful for your interest.

-          The authors sshould also provide data about muscle weakness or mobility since many surgeons require rapid recovery of muscle strength after surgery and early mobilization

A-4) We are grateful for your valuable suggestion. Firstly, it is important to note that this issue represents the most significant limitation of our study. This was explicitly stated in the section on limitations in the initial submission of our article. Furthermore, we would like to clarify that we have expanded our limitation in order to provide a more comprehensive understanding of this issue. It is evident that further studies are necessary to gain a deeper insight into this subject. Although we have provided a detailed explanation of this situation in the limitation section of our study, we would like to reiterate in your presence that our orthopaedic surgery clinic applies postoperative immobilisation for at least 12 hours to patients who have undergone hip surgery. This is due to the surgical clinical approach and because we are anaesthetists, we are unable to intervene for reasons beyond our control. Consequently, it was not possible to perform an effective muscle strength assessment on the study patients. Although we did perform minor muscle movement for pain caused by movement in the postoperative PACU, we did not consider it appropriate to include this result in the muscle strength assessment. It is recommended that further studies be conducted on this subject. We would like to state that we have expanded the limitations section of our study in accordance with your valuable suggestion.

 Discussion:

-          While the blocks were performed under general anaesthesia it is not possible to evaluate the effectiveness oft he block. This should be included in the discussion section and should be discussed in the light oft he recent literature.

A-5) I would like to express my gratitude for your invaluable contributions. The data presented in the Discussion section are derived from the literature and relate to patients undergoing general anaesthesia. The Tulgar et al. study included the application of L-ESP block to patients undergoing general anaesthesia. In order to clarify the matter, we wish to state that we wrote separately that the study was performed on patients under general anaesthesia. It is also important to consider that the application of a peripheral block to a conscious patient can result in the development of injection-induced pain and subsequent anxiety. A detailed discussion of the application of peripheral blocks under general anaesthesia and mild sedation, with supporting literature, is presented in the discussion section. In response to your valuable suggestion, the discussion section has been expanded. We are grateful for your support. 

Reviewer 2 Report

Comments and Suggestions for Authors

Thank you for permitting me to review this mauyscript 

In this double blind study , the authors compared  postoperative analgesia in 3 groups  of hip arthroplasty and conclude peng block and spinae erector block improved postoperative anlgesia in this type of surgery 

here are my comments 

line 70-72 : the authors speak about of severe postoperative pain in hip surgery , however this surgery is not considered as having severe postoperative pain , indeed the authors themselves found an average of 21 mg  of morphine in the first postoperative day , I think the term severe can be deleted 

Its not clear if the postoperative pain was assessed blindely as the opioid management , please specify these issues 

Based on the results of the study may be the authors can also state postoperative analgesia was not different between the two blocks 

The authors did not evaluate dynamic pains , however the basis of hip surgery is to permit early mobilization may be the authors could discuss this issue 

Figure 3 and 4 need more statistical explanations 

Author Response

Dear Editor,

We thank the reviewers for their valuable comments on the manuscript, and we have edited the manuscript to address their concerns. We have highlighted in red all the changed places in the main text. We would like to state that all the reviewers' comments have been taken into account and the necessary corrections have been made.

We hope that the article is now suitable for publication.

Yours sincerely

Ali EYRÄ°CE, MD.

Department of Anesthesiology and Reanimation, Trakya University Medical Faculty, Edirne, Turkey

On behalf of all authors.

Reply to Reviewer-2:

Thank you for permitting me to review this mauyscript 

In this double blind study , the authors compared  postoperative analgesia in 3 groups  of hip arthroplasty and conclude peng block and spinae erector block improved postoperative anlgesia in this type of surgery 

here are my comments 

  • line 70-72 : the authors speak about of severe postoperative pain in hip surgery , however this surgery is not considered as having severe postoperative pain , indeed the authors themselves found an average of 21 mg  of morphine in the first postoperative day , I think the term severe can be deleted 

A-1) In response to your valuable suggestion, we have made the requisite changes to the specified part. I would like to express my gratitude for your assistance.

  • Its not clear if the postoperative pain was assessed blindely as the opioid management , please specify these issues 

A-2) I would like to express my gratitude for your invaluable contribution. In accordance with your recommendation, a more comprehensive description has been incorporated into the material and method section of our study.

  • Based on the results of the study may be the authors can also state postoperative analgesia was not different between the two blocks 

A-3) I would like to express my gratitude for your valuable suggestion. In light of your input, we have revised the conclusion section of our study.

  • The authors did not evaluate dynamic pains , however the basis of hip surgery is to permit early mobilization may be the authors could discuss this issue 

A-4) I would like to express my gratitude for your valuable suggestion. In accordance with your suggestion, we have expanded the discussion section and limitation section of our study to encompass this issue. Furthermore, it should be noted that the inability to effectively evaluate muscle strength in patients is due to the application procedure of the orthopaedic clinic. Furthermore, we would like to state that this issue has been included in the section on limitations.

  • Figure 3 and 4 need more statistical explanations 

A-5) I would like to express my gratitude for your invaluable contribution. In accordance with your recommendation, Figure 4 has been entirely revised and the significance values have been incorporated. As Figure 3 is a plot of change over time, a detailed explanation of this is provided in the conclusion. We are grateful for your constructive suggestions and valuable contributions.
